# The Labour Conditions and Health of Migrant Agricultural Workers in Spain: A Qualitative Study

**DOI:** 10.3390/healthcare13151877

**Published:** 2025-07-31

**Authors:** Vanesa Villa-Cordero, Amalia Sillero Sillero, María del Mar Pastor-Bravo, Iratxe Pérez-Urdiales, María del Mar Jiménez-Lasserrotte, Erica Briones-Vozmediano

**Affiliations:** 1Escola Universitària Gimbernat, Adscrita a la Universitat Autònoma de Barcelona (UAB), 08174 Sant Cugat, Spain; vanesa.villa@eug.es; 2Department of Nursing, University of Murcia, 30100 Murcia, Spain; 3Pascual Parrilla Murcian Institute of Biosanitary Research (IMIB), 30120 Murcia, Spain; 4Department of Nursing I, University of the Basque Country (UPV/EHU), 48013 Bilbao, Spain; iratxe.perez@ehu.eus; 5Biocruces Bizkaia Health Research Institute, 48903 Barakaldo, Spain; 6Department of Nursing, Physiotherapy, and Medicine, University of Almería, 04120 Almería, Spain; mjl095@ual.es; 7Department and Faculty of Nursing and Physiotherapy, University of Lleida, 25003 Lleida, Spain; erica.briones@udl.cat; 8Consolidated Research Group in Society, Health, Education and Culture (GESEC), University of Lleida, 25003 Lleida, Spain; 9Research Group in Healthcare (GRECS), Biomedical Research Institute of Lleida (IRB-Lleida), Josep Pfiarré Foundation, 25198 Lleida, Spain

**Keywords:** migrant agricultural workers, work disparities, occupational health, health inequities, working conditions, social determinants of health, qualitative research, Spain

## Abstract

**Background/Objectives**: Agricultural workers in Spain with a migratory background face challenging working and living conditions that significantly affect their health. This study aimed to explore how professionals in healthcare, social services, civil society organisations, and labour institutions perceive that the working conditions affect the physical health of this population. **Methods**: A qualitative descriptive study was conducted through 92 semi-structured interviews with professionals from six provinces in Spain. Data were analysed using thematic analysis following Braun and Clarke’s six-phase framework. Rigour was ensured through triangulation, independent coding, and interdisciplinary consensus. **Results**: Two overarching themes were identified: (1) the health consequences of workplace demands and environmental hazards, and (2) navigating health services such as sick leave and disability permits. These findings highlight how the impact of precarious working conditions and limited access to healthcare affect the physical health of migrant agricultural workers. **Conclusions**: The professionals interviewed described and relate precarious working conditions with adverse health outcomes among migrant agricultural workers. Their insights reveal the need for systemic reforms to enforce labour rights, ensure access to health services, and address the structural factors that contribute to exclusion and vulnerability.

## 1. Introduction

Agriculture represents up to 6% of Spain’s labour force [1] and is widely recognised as one of the most hazardous and demanding occupational sectors [2,3]. It consistently ranks among the top ten industries with the highest incidence of occupational illnesses (85 cases per 100,000 worker-years), particularly non-traumatic disorders linked to prolonged physical exertion and exposure to harmful environmental agents. Furthermore, it ranks third in occupational diseases attributed to biological agents [4].

Agricultural labour is closely associated with musculoskeletal, cardiovascular, and respiratory disorders, as well as an increased risk of zoonotic infections such as brucellosis, and parasitic or arboviral diseases [5]. Prolonged exposure to solar radiation and agrochemicals, especially pesticides and fertiliser, has been linked to heightened risks of neurological and oncological conditions, with agricultural workers showing disproportionately high rates of cancers affecting the central nervous system, lungs, prostate, and breast [6]. Concerning occupational injuries, the primary sector (including fishing) ranks second in Spain [7], with agriculture alone accounting for 8% of all fatal work-related incidents [1]. The seasonal and cyclical nature of agricultural work, marked by alternating periods of intensive activity and inactivity, has rendered it one of the sectors with the highest levels of job insecurity and temporary contracts [7,8]. As a result of these conditions and the increasing demands of productivity, the native Spanish workforce has been gradually replaced by migrant labourers over the past decades [9].

Labour migration remains a global phenomenon and a primary driver of cross-border mobility for millions of people [10]. Migrant agricultural workers (MAWs) are essential to the sustainability of agri-food systems, comprising up to 70% of the agricultural workforce in Spain [10,11]. An estimated 30% of MAWs in Europe are employed in irregular situations, without formal contracts or documentation, which exacerbates their social and legal vulnerability [12].

While recent trends show an increase in the feminisation of migration [13], only 25% of formal MAW contracts are held by women [14]. Gender plays a key role in shaping labour roles and inequalities: men are more likely to obtain relatively stable positions in manual or manufacturing work, while women more often face instability and are relegated to temporary packaging roles. In Spain, the predominant profile of seasonal MAWs is a 36.9-year-old male, with 26.5% of these workers lacking legal residency status [8].

MAWs endure particularly strenuous labour conditions. Their work is physically demanding and often performed under extreme climatic conditions. Their weekly schedules exceed 50 h, and have minimal or no rest days [15]. Wages frequently fall below legal thresholds, payments are often informal, and overtime remains uncompensated [7,8,16].

Living conditions are equally precarious. Many MAWs reside in overcrowded, informal dwellings near agricultural sites, often lacking essential services such as potable water and electricity [17,18].

Access to healthcare is particularly limited for undocumented MAWs, who face multiple barriers, including administrative constraints, linguistic difficulties, and legal exclusions [19,20]. These obstacles hinder the prevention, early detection, and effective treatment of health conditions [21], including infectious diseases such as tuberculosis or parasitic infections [22], which increase the risk of outbreaks within and beyond the MAW population. Broader social determinants, such as socioeconomic inequality, occupational exclusion, and systemic marginalisation, further intensify the likelihood of syndemic health outcomes [23].

Previous research has also underscored the impact of water insecurity and poor hygiene on the health of MAWs in Spain, highlighting how substandard access to safe drinking water and sanitation correlates with increased risk of illness [24]. However, little attention has been paid to working conditions, distinct from housing or water-related factors, as direct social determinants of health. Few studies have considered the views of professionals, whose perspectives are essential for identifying occupational risks and informing preventive strategies. To address this gap, this study asks the following: How do professionals in the social and healthcare sectors perceive the consequences of the socio-labour conditions of MAWs on their physical health? The aim was to explore the consequences of working conditions on the physical health of MAWs from the perspective of professionals involved in their care and support, providing evidence to inform preventive and rights-based public health strategies.

## 2. Materials and Methods

### 2.1. Study Design

This descriptive exploratory qualitative study is part of the first stage of a broader mixed-methods research project to assess the impact of occupational precarity and social exclusion on the health and well-being of migrant men and women employed in Spain’s agricultural sector [25]. The study employed a qualitative descriptive design, as described by Kim et al. [26], which is particularly suitable for exploring understudied or complex phenomena using straightforward, data-driven interpretations. The study sought to gather the knowledge and perceptions of professionals in health, social care, NGOs (non-governmental organisations), and trade unions regarding the health implications of MAWs’ working and living conditions. Semi-structured individual interviews were conducted between 2021 and 2022 in six Spanish provinces across four autonomous regions with high concentrations of seasonal MAWs [12]. The study adhered to the Consolidated Criteria for Reporting Qualitative Research (COREQ) guidelines [27].

### 2.2. Participants

Participants were recruited using purposive sampling in each province, initially by identifying key informants through institutional contacts and publicly available listings, and subsequently through snowball sampling. Eligible participants were professionals with at least six months of experience working with migrant agricultural populations, ensuring sufficient familiarity with their working and living conditions. This inclusion criterion was established to guarantee that participants could provide informed insights into occupational and health-related issues. The sample included healthcare providers, social workers, NGO staff, cultural mediators, and union representatives, whose direct involvement in healthcare, social support, and labour rights protection made them key informants for identifying occupational risks and health vulnerabilities among MAWs. A total of 92 participants (55 women and 37 men) were recruited, including professionals from the healthcare sector (*n* = 17), NGOs and civil society organisations (*n* = 18), social services (*n* = 24), and labour unions (*n* = 14), as well as cultural mediators (*n* = 8), COVID-19 contact tracers (*n* = 2), political representatives (*n* = 7), and businesspersons (*n* = 2). No one withdrew during the research. The interviews lasted approximately 60 min on average. They were conducted in six provinces across four autonomous regions with high concentrations of MAWs: Andalucia (*n* = 31), La Rioja (*n* = 20), Lleida (*n* = 24), and Murcia (*n* = 17) (see Table 1).

### 2.3. Data Collection

Personal semi-structured interviews were conducted with 92 participants until data saturation was achieved, at which point no new concepts emerged. The research team designed an interview guide based on previous literature and their own experience. It included open-ended questions grouped into three categories: (1) living conditions, (2) working conditions, and (3) their health. The guide was piloted in the first interviews to ensure clarity and adaptability, and is provided as Appendix A for transparency.

Most interviews were conducted face-to-face at participants’ workplaces or public facilities, while others were conducted via video conferencing, depending on the participants’ availability and convenience. All interviews were recorded with prior consent, and transcribed verbatim.

### 2.4. Data Analysis

Interview transcripts were imported into ATLAS. Ti Web (version 9; ATLAS.ti Scientific Software Development GmbH, Berlin, Germany) to facilitate data organisation and thematic exploration. An inductive thematic analysis was performed based on the six-phase method outlined by Braun and Clarke [28], comprising (1) data familiarisation, (2) initial code generation, (3) theme identification, (4) theme review, (5) theme definition and naming, and (6) report production. This framework was selected for its flexibility and appropriateness for capturing nuanced participant experiences in applied health contexts.

A preliminary codebook was developed from the interview guide. Four researchers independently reviewed the transcripts, identifying meaning units (sentences or paragraphs) related to MAWs’ living, working, and health conditions. These were coded, grouped thematically, and organised into conceptually related families. Although the primary focus was on working conditions and health, references to living conditions were retained during analysis because participants consistently linked them to occupational health outcomes (e.g., lack of rest, poor housing, and limited recovery between shifts).

Initial codes were refined and consolidated through iterative analysis and team discussions into emerging subcategories and overarching themes. Regular consensus meetings ensured interpretative consistency and analytical coherence across coders. Including diverse disciplinary perspectives and using illustrative participant quotations further strengthened the study’s credibility and transparency.

### 2.5. Ethical Considerations

This study adhered to the ethical standards outlined in the Declaration of Helsinki [29]. It complied with the General Data Protection Regulation (EU 2016/679) and Spain’s Organic Law 3/2018 on the Protection of Personal Data. Ethical approval was obtained from the Healthcare Ethics Committee of the Arnau of Vilanova Hospital in Lleida in 2021 (CEIC-2459). All participants received detailed information about the study aims and procedures, and gave written and verbal informed consent before participating. Confidentiality and anonymity were ensured throughout the study. Each participant was assigned a unique identifier during transcription, and all quotations were disidentified to prevent recognition.

### 2.6. Rigour and Reflexivity

Methodological rigour was ensured by following the criteria developed by Lincoln and Guba [30], which included credibility, transferability, and confirmability. Credibility was reinforced through the triangulation of coding and interpretation among five researchers, as well as the inclusion of verbatim quotes to support the thematic analysis. Transferability was addressed by providing detailed descriptions of the context and participant diversity. Confirmability was supported by the research team’s extensive experience in qualitative inquiry and by maintaining an audit trail of analytic decisions throughout the research process. Reflexivity was promoted through regular team discussions in which the researchers critically reflected on their roles, assumptions, and interactions with the data. The multidisciplinary composition of the team, comprising professionals with backgrounds in nursing, sociology, and public health, provided complementary perspectives. At the same time, their prior experience with migrant health issues facilitated contextual understanding but also required conscious effort to minimise potential biases. This reflexive approach enriched the interpretative process and enhanced the credibility of the findings.

## 3. Results

Niney-two professionals participated in the study, comprising 60% women (*n* = 55) and 40% men (*n* = 37). The mean age was 43.7 years (SD = 8.7), with women averaging 42.3 years (SD = 8.2) and men 45.5 years (SD = 9.1. The most represented professional roles were social workers (26%), NGO staff (20%), healthcare professionals (18%), and union representatives (15%) (see Table 1 for complete details).

### Thematic Analysis

The thematic analysis yielded two main themes: (1) the health consequences of workplace demands and environmental hazards, and (2) navigating health services including sick leaves and disability permits. Each theme encompasses several subthemes that illustrate how the working conditions of MAWs impact their physical health (Table 2).

Below, we present the key findings and participant narratives.

Theme 1: The health consequences of workplace demands and environmental hazards

This theme explored how the physical demands and environmental exposures associated with agricultural work negatively affect the health of MAWs. Three subthemes emerged:

1.1. Subtheme: Agricultural Work Demands

Professionals described agricultural work as extremely strenuous, requiring repetitive and physically demanding movements:

*“Strawberry picking is brutal; they have to stay crouched the whole time*”.
*(P83, woman, NGO Health Technician, Huelva)*


*“It’s an intensely physical job, especially the grape harvest”*.
*(P22, man, NGO Social Worker, La Rioja)*


They were also defined as hard due to the pressure to meet productivity targets, exacerbating stress and physical exhaustion:

*“You have to complete ten rows in two hours. If you fall behind, they scold you and rush you, saying, you’re going too slow, we have to do 500 rows per hour.”*.
*(P79, woman, Trade Union, Murcia)*


Participants often complained that employers do not respect the number of working hours or schedules. *“The law says 12 hours of rest between shifts, but many get only 5 or 6/…/Employers do not care—it’s about getting the product out […] It feels like slavery is back. Starting work at 3 a.m., returning exhausted—of course it affects their health” (P56, man, Trade Union, Murcia).*

Therefore, frequently, MAWs endure extreme fatigue, working long hours while sleeping outdoors or in substandard housing:

*“Ten-hour days, sleeping outdoors, and still working the next day—there’s no proper recovery”*.
*(P22, man, NGO Social Worker, La Rioja)*


1.2. Subtheme: Environmental Health Hazards and Risk Prevention

Participants explained how MAWs often work in harsh weather conditions, which are incompatible with health. Therefore, to avoid the most harmful hours or heat peaks, work schedules are adapted:

*“In summer, they start at 5 a.m. and finish by 11 due to greenhouse temperatures exceeding 56 °C”*.
*(P39, woman, Town Council Social Worker, Almería)*


Related to extreme temperatures, on the one hand, exposure to cold temperatures was linked to respiratory issues:

*“Many colds and coughs, especially due to insufficient clothing in winter”*.
*(P6, woman, NGO Nurse, Almería)*


On the other hand, exposure to extreme heat during the summer was linked to symptoms compatible with heatstroke, according to participants:

*“Temperatures inside greenhouses can reach 45 °C”*.
*(P83, woman, Health Technician NGO, Huelva)*


*“They get weak from the heat—classic heatstroke symptoms”*.
*(P74, man, Trade Union, Murcia)*


In addition, professionals identified dermatological problems in MAWs related to exposure to sun radiation and other natural risks, such as insect bites.

*“Plenty of skin issues—rashes, sunburns, and allergies”*.
*(P39, woman, Town Council Social Worker, Almería)*


*“They showed up covered in bites and eczema from fieldwork”*.
*(P29, woman, NGO Sociologist, Huelva)*


*“Mosquito bites often lead to wounds”*.
*(P19, man, Town Council Mediator, Lleida)*


Regarding risk prevention, employers often fail to provide workers with the necessary equipment and protective clothing to perform their duties safely, or do not provide them with access to drinking water.

*“They wear sandals even in November. It’s freezing, and they’re not properly clothed”*.
*(P66, woman, NGO Social Worker, Jaén)*


*“I’ve seen employers say: ‘Bring your water from home’—terrible conditions”*.
*(P35, man, Trade Union, Murcia)*


Another health risk identified by participants was the use of pesticides, including spraying. If it occurs during working hours, workers are exposed to pesticides without adequate protection:

*“You see them spraying while workers are nearby without any protection”*.
*(P77, man, Trade Union, Murcia)*


*“Sulphur is sprayed even when people are working—it’s extremely irritating/…/Some are allergic to Sulphur, and they’re not provided proper masks”*.
*(P74, man, Trade Union, Murcia)*


Despite not being considered work-related accidents, traffic accidents were described as common when MAWs commute to work, driving a bicycle or in overcrowded vans on narrow and dark roads in rural environments:

*“Many are hit while biking to work on narrow roads. Some ended up in a vegetative state”*.
*(P39, woman, Town Council Social Worker, Almería)*


*“Working from 5 a.m. to 8 p.m., then biking home—it’s easy to make mistakes and crash”*.
*(P77, man, Trade Union, Murcia)*


1.3. Subtheme: Health Consequences of Agricultural Work

Participants explained how systemic labour instability and inadequate protection mechanisms lead to poor health outcomes. For example, they identified carpal tunnel, circulatory system problems, open wounds, and back issues as common among MAWs’ occupational diseases, due to being intensively picking the products, standing all day in the packaging factories and using sharp objects to cut the vegetables and fruits:

*“This hospital has the highest rate of carpal tunnel surgeries, varicose veins, back issues—standing all day, lifting weights/…/Almost every woman in the packaging area gets carpal tunnel syndrome”*.
*(P39, woman, Town Council Social Worker, Almería)*


*“Falls, hernias, back injuries from lifting crates or slipping on lettuce leaves/…/Knife cuts are common—especially during broccoli or lettuce harvesting”*.
*(P56, man, Trade Union, Murcia)*


In this sense, they explained that being exposed to continued adverse working conditions during the agricultural campaigns can lead to present symptoms of severe asthenia, significant physical deterioration and premature ageing, as they see in MAWs:

*“They age fast—it’s not just how they look, it’s their whole health that collapses”*.
*(P31, woman, Trade Union, La Rioja)*


*“Some are retired by 45 due to spine damage—they can’t enjoy retirement because their bodies are wrecked”*.
*(P56, man, Trade Union, Murcia)*


*“They lose weight, can barely stand. It’s pure exhaustion”*.
*(P19, man, Town Council Mediator, Lleida)*


Theme 2: Navigating health services: sick leave and disability permits

This theme describes the challenges of receiving appropriate attention in a healthcare centre when an accident occurs or health problems related to work arise. Three subthemes emerged:

2.1. Subtheme: Occupational Health and Health Service Coverage

The interviewed professionals identified some problems regarding the health services coverage for MAWs. Firstly, participants reported that occupational health screenings provided by mutual insurance companies are often superficial or mistrusted:

*“They asked for medical certificates, but the exams were verbal— ‘Any health issues?’ ‘No.’ And that’s it, they sign it”*.
*(P39, woman, Town Council Social Worker, Almería)*


Secondly, MAWs can be reluctant to attend health services for preventive uses:

*“Many of them refuse to attend the screenings.”*.
*(P26, man, Secretary General Trade Union, La Rioja)*


Participants noted that MAWs often downplay symptoms and avoid resting or seeking care to avoid missing work, as they fear that if they do not work, they will not receive a salary.

*“She came in at 5 a.m. with a migraine, just wanted oxygen quickly so she could go to work”*.
*(P82, woman, Doctor Health Sector, Huelva)*


*“He had a broken leg and still wanted to work”*.
*(P23, woman, Trade Union, La Rioja)*


*“They care about the job—they need to send money home. If they feel better, they work, no matter the diagnosis. Working means receiving money, otherwise you do not receive it.”*.
*(P5, man, Health Sector Nurse, Lleida)*


2.2. Subtheme: Work-Related Accidents

The interviewed professionals identified some bad practices among employers regarding their reaction to work-related accidents. For example, when instructing workers to pretend the accident occurred during their spare time while attending health services.

*“Some injuries are never reported—the employer just tells them to lie”*.
*(P11, woman, Professor and journalist in NGO, Lleida)*


*“If the injury isn’t serious, they’re told to say it happened at home—even when pruning or using tools”*.
*(P77, man, Trade Union, Murcia)*


Alternatively, even regarding cases when employers abandoned an injured worker in front of the hospital to avoid any responsibility, participants commented the following:

*“Workers said they were dropped off (by the employer) at the hospital after fainting or being injured—and then never called back or paid”*.
*(P73, woman, NGO psychologist, Murcia)*


*“Employers only act when the condition is severe, life-threatening even”*.
*(P29, woman, NGO Sociologist, Huelva)*


Moreover, when work-related accidents occur, the lack of job regularisation often makes it difficult to benefit from the health service coverage:

*“In normal conditions, they would receive disability benefits, but without papers or contributions, they fall through the cracks”*.
*(P53, woman, Social Worker Health Sector, La Rioja)*


*“Most are young adult men with back pain or other issues, often stemming from unreported work injuries”*.
*(E52, woman, Social Worker, Town Council)*


2.3. Subtheme: Challenges with Temporary or Permanent Disabilities

Due to a lack of protection and labour regulation for MAWs, there are employers who, instead of accepting a sick leave request from a worker, terminate the employee’s contract. In cases of permanent disability or severe illness, undocumented MAWs are left unprotected without a disability pension and people supporting them:

*“A fracture? That’s a big deal. If you go on leave, you do not get paid, and the boss wants to fire you quickly”*.
*(P32, man, Mediator Health Sector, Lleida)*


*“If they can’t return home, they just drift around here, unable to work and without support”*.
*(P53, woman, Social Worker Health Sector, La Rioja)*


To better illustrate these findings, a conceptual diagram (Figure 1) has been included. This figure summarises the interrelated factors identified in the thematic analysis, showing how specific labour conditions (e.g., long working hours, poor ergonomics, and exposure to chemicals) and structural vulnerabilities (e.g., precarious employment, inadequate housing, and limited healthcare access) interact to shape health outcomes among MAWs, including musculoskeletal disorders, respiratory problems, and psychological distress.

## 4. Discussion

This study highlights how social and healthcare professionals perceive the precarious labour conditions of MAWs in Spain as a critical determinant of their physical health. According to participants, these workers face a convergence of occupational hazards and structural vulnerabilities stemming from their migrant status that significantly deteriorate their well-being. They explained how the absence of labour protections, coupled with social exclusion, increases the likelihood of exposure to injury, chronic illness, and psychosocial stress.

These findings reflect the work disparities and inequities faced by MAWs. Such disparities are deeply structural, arising from legal precarity, temporary and informal contracts, and social exclusion, which collectively intensify their health vulnerabilities. Addressing these inequities requires rights-based, equity-oriented interventions that go beyond short-term measures [19].

Our results are consistent with prior Spanish and international research, which has reported a high prevalence of respiratory and dermatological conditions among agricultural workers, often associated with pesticide exposure and ultraviolet radiation [5,7,17,31]. This alignment with previous studies reinforces the persistent nature of these health risks. In addition, the frequent occurrence of musculoskeletal disorders associated with repetitive tasks and poor ergonomic conditions in this population has been widely documented [32]. The present findings further confirm it.

The cumulative impact of occupational and environmental exposures in agriculture, encompassing risks from zoonotic agents and long-term physical strain, has also been substantiated by European regulatory reports and scientific reviews [33]. Our participants’ accounts of toxic pesticide use without adequate protective equipment for the workers echo the health warnings issued by the European Food Safety Authority, particularly regarding the chronic and acute toxicities resulting from unregulated agrochemical use [33,34]. Moreover, previous scientific evidence has demonstrated that occupational exposure to pesticides is strongly associated with respiratory disorders, such as asthma, chronic bronchitis, and reduced lung function, among agricultural workers [35,36,37]. These findings are directly mirrored in the narratives collected in this study.

The discussion of fear of dismissal or deportation, which discourages workers from reporting symptoms or seeking healthcare, is not new in the literature. Previous studies have documented the health consequences of legal precarity, irregular migration status, and lack of occupational training [38], and our results provide qualitative depth to this pattern. Similarly, the economic hardship and disillusionment described by MAWs after migration are consistent with national reports highlighting the gap between migrant workers’ aspirations and the realities of low-wage sectors such as agriculture [39].

Regarding healthcare access, our findings align with existing literature that underscores the vulnerability of undocumented workers. As previously reported [20,38], the absence of formal contracts and fear of institutional contact limit many MAWs’ ability to access care or adhere to treatments. The present study extends this evidence by revealing how these barriers are perceived and managed on a daily basis by frontline professionals.

The interplay of precarious employment, inadequate housing, and exclusion from healthcare exposes MAWs to intersecting vulnerabilities, a phenomenon well described by Singer’s syndemics framework [23]. For MAWs, this syndemic involves cardiovascular, respiratory, and infectious diseases exacerbated by chronic stress, environmental exposures, and social exclusion, as documented in prior international analyses [40,41] and further substantiated by our qualitative findings.

The participants‘ accounts of overcrowding, poor sanitation, and inadequate water access in informal settlements support the association—previously established by public health reports and NGO alerts—between substandard housing, water insecurity, and an increased risk of infectious disease [42]. Housing-related health challenges for migrant agricultural workers are a persistent and widely acknowledged issue. The negative impact of overcrowding, poor ventilation, and inadequate sanitation in employer-provided housing on respiratory health, infection risk, and mental health deterioration is reflected in the lived experiences shared by our participants [43].

This study further aligns with recent European and North American investigations that document ongoing deficits in occupational safety, legal protection, and access to healthcare among seasonal migrant workers [44,45]. Despite the existence of specific reforms and humanitarian programs, the professionals interviewed here generally perceived their impact as limited. International evaluations continue to highlight the persistent challenges in enforcing labour rights, particularly for undocumented workers [10,44]. Furthermore, our findings reinforce recent calls in the literature for improved cross-sectoral coordination and structural responses to mitigate these vulnerabilities, particularly during crises such as the COVID-19 pandemic [45].

By incorporating diverse professional perspectives from multiple Spanish regions, this study offers a comprehensive understanding of how structural determinants influence both health and access to healthcare among MAWs. Considering these findings, there is a clear need to implement structural interventions that extend beyond short-term aid. As previously recommended [9], policies should focus on strengthening labour rights enforcement, improving housing conditions, guaranteeing universal access to healthcare, and establishing regularisation pathways for undocumented workers. Only through such multifaceted, rights-based approaches can the cycle of exclusion and health inequity be effectively addressed.

### 4.1. Limitations and Strengths

This study has several limitations. First, the findings are based exclusively on the perspectives of professionals involved in the care and support of MAWs. While this provides valuable insights into systemic issues and health consequences that may go unnoticed by the workers themselves, it does not capture the direct voices and lived experiences of MAWs. Incorporating the perspectives of the workers is essential for achieving a more nuanced and grounded understanding of their health-related challenges. This limitation is already being addressed in the next phase of the AGROMISALUD project, which will directly explore the narratives of MAWs [25].

Second, the fieldwork was conducted in a post-COVID-19 context. The extent to which pandemic-related health measures (e.g., improved housing, sanitation, or employer oversight) had a temporary or lasting effect on the working and living conditions of MAWs remains unclear. Future research should assess the continuity and long-term impact of these changes.

Despite these limitations, the study has several strengths. Notably, it draws on a large and diverse sample of 92 informants, including professionals from healthcare, social services, NGOs, and trade unions. This heterogeneity enhances the credibility and transferability of the findings, as it reflects a broad range of experiences and regional contexts across key agricultural provinces in Spain. The multisectoral perspective also enables the triangulation of viewpoints, thereby contributing to the depth and reliability of the analysis.

Moreover, by incorporating the perspectives of professionals directly involved in the care and support of MAWs—an approach rarely addressed in previous research—this study identifies health problems not always self-recognised by MAWs, such as early signs of physical deterioration or psychosocial distress. By highlighting the clinical and structural dimensions of their vulnerability, this research supports the development of more contextualised and effective public health strategies aimed at reducing occupational and social health inequities among migrant agricultural workers.

### 4.2. Implications for Practice

The findings of this study highlight the urgent need to move away from the normalisation of precariousness and vulnerability in the agricultural sector. Ensuring health equity for MAWs requires cross-sectoral collaboration to address structural determinants, such as insecure employment, unsafe workplaces, and barriers to healthcare, that undermine both physical and mental health. Strategies should include the enforcement of labour protections, universal access to healthcare regardless of legal status, and the promotion of occupational health literacy for both workers and employers. Recognising MAWs as rights holders entitled to protection, fair working conditions, and healthcare is fundamental for advancing public health and social justice in the agriculture sector.

## 5. Conclusions

This study provides multi-professional evidence that precarious socio-labour conditions, characterised by harsh employment settings, inadequate housing, and restricted healthcare access, are widely perceived by key stakeholders as significant determinants of ill health among MAWs in Spain. The broad regional and institutional scope of the sample enables an in-depth analysis of systemic barriers, revealing risks and vulnerabilities that often remain unaddressed in official data and policy discussions.

Beyond describing these challenges, the study calls for immediate, coordinated policy action to protect the health and rights of MAWs. This includes strict enforcement of labour standards, improvements in the accessibility and quality of healthcare, and targeted educational interventions to raise awareness among workers and employers about occupational risks. Such measures should be embedded in comprehensive, rights-based frameworks to ensure that all agricultural workers are afforded dignity, safety, and practical support.

Future research should focus on incorporating the lived experiences and perspectives of MAWs themselves, thereby enriching the understanding of their specific needs and resilience strategies. It is also crucial to assess the sustainability and impact of post-pandemic policy changes on working and living conditions within the sector.

Ultimately, this research supports the development of context-sensitive public health strategies that address immediate risks and contribute to long-term structural change. Advancing health equity for MAWs is not only a matter of occupational safety, but also of upholding fundamental human rights in one of the most vulnerable sectors of the workforce.

## Figures and Tables

**Figure 1 healthcare-13-01877-f001:**
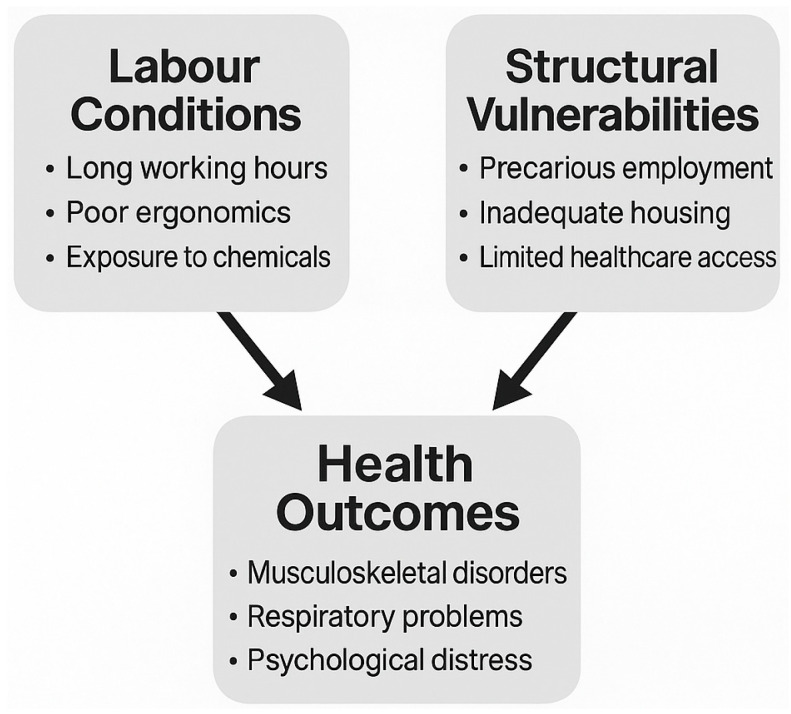
Conceptual diagram.

**Table 1 healthcare-13-01877-t001:** Sociodemographic characteristics of the participants (*n* = 92)**.**

Variable	Category	Women (*n*)	Men (*n*)	Total (*n*)	%
Professional Role	Healthcare professionals	10	7	17	18%
	Social workers	16	8	24	26%
	Cultural mediators	5	3	8	9%
	NGO staff	13	5	18	20%
	COVID-19 contact tracers	2	0	2	2%
	Union representatives	4	10	14	15%
	Political officials	3	4	7	8%
	Businesspeople	2	0	2	2%
Age Group	18–25	1	0	1	1%
	26–40	21	9	30	33%
	41–50	20	16	36	39%
	51–65	13	12	25	27%
	Mean age (SD)	43.2 (10.2)	46.5 (9.5)	44.6 (10.0)	
Province	Andalucia	18	13	31	34%
	La Rioja	11	9	20	22%
	Lleida	15	9	24	26%
	Murcia	11	6	17	18%
Total by Sex		55	37	92	100%

**Table 2 healthcare-13-01877-t002:** Themes and subthemes.

Theme	Subthem
1 The health consequences of workplace demands and environmental hazards	1.1: Agricultural work demands 1.2: Environmental health hazards and risk prevention 1.3: Health consequences of agricultural work
2 Navigating health services: sick leave and disability permits	2.1: Occupational health and health services coverage 2.2: Work-related accidents 2.3: Challenges with temporary or permanent disabilities

## Data Availability

The datasets generated and/or analysed during the current study are not publicly available but are available from the corresponding author upon reasonable requests.

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
