# Peer review of "The Labour Conditions and Health of Migrant Agricultural Workers in Spain: A Qualitative Study"

_healthcare, 2025, doi:10.3390/healthcare13151877_

Round 1
Reviewer 1 Report
Comments and Suggestions for Authors
See attached report.

Author Response
Dear Reviewer 1,
Please find attached the point-by-point response document to your comments. We sincerely appreciate your valuable feedback, which has significantly contributed to improving the quality and clarity of our manuscript.
Best regards,

Reviewer 2 Report
Comments and Suggestions for Authors
The study investigates how professionals in healthcare, social services, NGOs, and labour institutions perceive the impact of socio-labour conditions on the physical health of migrant agricultural workers (MAWs) in Spain. It aims to understand health's structural and occupational determinants through 92 interviews conducted across six Spanish provinces.
The topic is both original and highly relevant. While many studies address migrant health or labour conditions separately, this study combines both through a qualitative, multisectoral lens. It fills an essential gap in the literature by highlighting institutional and professional perspectives, which are often overlooked in favour of self-reports or epidemiological data.
This article offers new empirical insights into how precarious working conditions, limited labour protections, and legal status intersect to shape the health outcomes of MAWs. It strengthens the existing literature by applying thematic analysis to a large and diverse set of professional stakeholders and highlighting systemic barriers not easily captured in quantitative research.
The conclusions are well-supported by the data and discussion. The article effectively links working conditions to health outcomes and integrates these findings within a broader socio-political framework. The inclusion of participant quotations enhances both the narrative and analytical coherence.
The references are comprehensive, relevant, and up to date. The authors engage well with national and international literature, including institutional reports and peer-reviewed sources.
Table 1 is informative and presents the participant characteristics. Table 2 appropriately organises the themes and subthemes.
Here are some specific improvements that the authors should consider
- The qualitative design is appropriate and well-executed, clearly adhering to COREQ guidelines.
- The reflexivity of the research team could be elaborated further, particularly regarding their positionality in the interpretation of data.
- While the study’s limitations are acknowledged, the absence of MAW voices in this phase should be more explicitly framed as a methodological constraint.
- The interview guide, which was pilot-tested, could be included as supplementary material for transparency.
- The authors might consider including a conceptual figure or diagram to summarise the relationships between labour conditions, health outcomes, and structural determinants.
As points for improvement, a careful language review is recommended to correct minor grammatical and stylistic inconsistencies, which could be adjusted to enhance reading fluency while not compromising the overall understanding of the text. It would also be appropriate to include a brief additional reflection on the researchers’ positionality about the study object and the interviewed participants, thereby reinforcing the reflexive dimension of the research process.
The study holds significant scientific merit, contributing to advancing knowledge in occupational health, social vulnerability, and the rights of migrant workers in European rural contexts. With the suggested minor adjustments, the manuscript will be in excellent condition for publication.
Comments on the Quality of English Language
The manuscript is generally well written, with a clear structure and appropriate academic tone. However, minor grammatical inconsistencies and stylistic issues are present throughout the text. A careful language revision by a native or professional English editor is recommended to enhance fluency, ensure consistency, and improve the overall readability of the manuscript. These refinements will further strengthen the clarity and impact of the study's contributions.
Author Response
Dear Reviewer 2,
Please find attached the point-by-point response document to your comments. We sincerely appreciate your valuable feedback, which has significantly contributed to improving the quality and clarity of our manuscript.
Best regards,

Round 2
Reviewer 1 Report
Comments and Suggestions for Authors
Dear Authors, thank you for your willingness to consider my feedback and revise the manuscript accordingly. I believe the paper has improved and now deserves publication. I have two minor issues to highlight: (1) you should probably check the order of your reference list; (2) the issue of "hazards and vulnerabilities stemming from migrant status versus agricultural work in general" has not been fully resolved. However, it probably can't be resolved with your data. This might be an interesting topic for future research. Good luck with your work!